# PHYSICS-BASED DECODING IMPROVES MAGNETIC RESONANCE FINGERPRINTING

## ABSTRACT

Magnetic Resonance Fingerprinting (MRF) is a promising paradigm to perform fast quantitative Magnetic Resonance Imaging (QMRI). However, existing MRF methods suffer from slow imaging speeds and poor generalization performance on radio frequency pulse sequences generated in various scenarios. To address these issues, we propose a novel MRI physics-informed learning approach for MRF. The proposed approach adopts a supervised encoder-decoder framework, where the encoder predicts the target tissue properties and the decoder reconstructs the inputs using the MRF physics. Specifically, the encoder embeds high-dimensional MRF time sequences into a low-dimensional tissue property space, while the decoder exploits an MRI physics model to reconstruct the input signals using the predicted tissue properties and associated MRI settings. It allows to learn better representations by integrating a fast and differentiable MRI physics model as the physics-informed regularization. The physics-based decoder improves the generalization performance and uniform stability by a considerable margin in practical out-of-distribution settings. Extensive experiments verified the effectiveness of the proposed physics-based decoding and achieved state-of-the-art performance on tissue property estimation.

## 1 INTRODUCTION

Quantitative Magnetic Resonance Imaging (QMRI) is used to identify tissue's intrinsic properties, such as the spin-lattice magnetic relaxation time (T1), the spin–spin magnetic relaxation time (T2), and other physical properties. Compared to conventional weighted (qualitative) MRI that focuses on tissue's contrast of brightness and darkness, QMRI reveals tissue's intrinsic properties with quantitative values and associated physical interpretations. Since different tissues are characterized by their distinct properties values, QMRI shows great potential to reduce subjectivity, with advantages in many areas including diagnosis, tissue characterization, investigation of disease pathologies, etc. [3, 30, 61].

Magnetic Resonance Fingerprinting (MRF) provides an alternative QMRI framework to achieve multi-property quantification simultaneously [43]. Given a pseudo-random radio frequency (RF) pulse sequence, a distinct magnetic response – a.k.a. fingerprint, signature, or signal evolution – from each specific tissue is observed and then used to predict the target tissue properties. Therefore, multi-property quantification boils down to an inverse problem that aims to infer underlying tissue properties from the magnetic responses.

Various approaches have been developed to solve the MRF problem, using model-based techniques, e.g. dictionary matching (DM), compressive sensing, as well as learning-based / data-driven techniques [6, 8, 9, 11, 15, 16, 18, 25, 38, 43, 46, 47, 50, 55, 57, 58]. In spite of good performance in particular situations, they rarely take the MRI dynamics into consideration. This can cause reduced robustness and generalizability to potential data shifts occurred in practical scenarios with serious negative consequences. For example, the T1 and T2 value range and distribution are patient-specific and subject to pathological tissue types, development phase and other factors, which may cause label shift. In addition, as specific RF settings can often be applied to different situations, hospitals, and MRI instruments, MRF models are naturally expected to be able to handle such varied cases and be generalized to different RF settings. Motivated by these issues, we aim to develop a new MRF

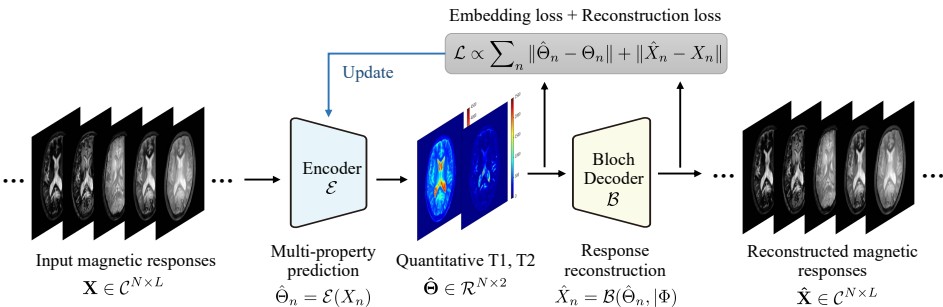

Figure 1: Diagram of the proposed BlochNet, a physics-based encoder-decoder for MRF. BlochNet adopts a supervised encoder-decoder framework where the encoder solves an inverse problem that predicts tissue properties from input magnetic responses while the decoder leverages an Bloch equation based MRI physics model to reconstructs the input responses from the estimated tissue properties. Such design helps the encoder capture generalizable mapping effectively with the aid of physics-based feedback from the Bloch decoder.

approach that combines the benefits of both model-based and learning-based techniques to achieve superior robustness and generalizability.

In this work, we propose a physics-based model, called *BlochNet*, to learn generalizable representations for MRF, as shown in Fig. 1. BlochNet adopts a supervised encoder-decoder framework where the encoder solves the inverse problem that predicts tissue properties from input magnetic responses, while the decoder leverages a Bloch equation based MRI physics model to reconstructs the input responses from the estimated tissue properties. The Bloch equations, a.k.a. equations of motion of nuclear magnetization, are a set of ordinary differential equations that model the magnetization dynamics and calculate the nuclear magnetization as a function of relaxation times T1 and T2 [4]. They enable the formulation of the magnetic response of a tissue with specific intrinsic properties T1, T2 under varied magnetic field (we provide more details in Section 3.3 and Appendix). The physics-based decoder acts as a causal regularization to perform supervised reconstruction to the input magnetic responses from estimated tissue properties as well as the associated pseudo-random excitation pulse sequence, such as repetition time (TR), time of echo (TE), and radio frequency flip angle (FA) over time. The rationale underlying the design is that domain knowledge such as physics principles can help to reduce the solution space of an inverse problem by applying additional constraints. This contributes to finding a better solution for an (ill-posed) inverse problem [31, 35, 41, 45]. Our results verify that the proposed model exhibits improved generalization performance from synthetic to real MRF data in two different out-of-distribution (OOD) settings, where the test data is generated from seen and unseen RF pulses in the first and second setting, respectively.

Our major contributions include:

- Our proposed method BlochNet is the first approach to incorporate physics-informed learning to solve MRF problems, where the Bloch equation-based MRI physics model serves as the decoder and guides the encoder to learn underlying robust representations by providing useful feedback for searching possible tissue properties.
- Compared to earlier methods, BlochNet shows consistently better generalization performance across synthetic, phantom and real MRF data, and across different types of RF pulse sequences. This demonstrates well the benefits of physics-based decoding in MRF in practical scenarios.
- We provide a fast and end-to-end solvable MRI physics model that can be used directly as a differentiable module in neural networks (*e.g.*, it acts as a decoder in BlochNet).

## 2 RELATED WORKS AND PRELIMINARIES

### 2.1 MODEL-BASED AND LEARNING-BASED MRF APPROACHES

As tissue properties result in magnetic responses through the MRI dynamics, quantifying tissue's properties via QMRI/MRF is an typical anti-causal task. The core idea of MRF is based on fact that

for each specific tissue, a pseudo-random pulse sequence leads to an unique magnetic response (i.e. magnetizition along the temporal dimension) which can serve as an identifiable signal signature, analogue to a "fingerprint", for the corresponding tissue. Once the magnetic responses are obtained, estimation of tissue properties from responses reduces to a pattern recognition problem. In the original MRF work [43], this is addressed via dictionary matching (DM) which finds the best matching entry in a pre-computed dictionary for each inquiry magnetic response. Accordingly, the best matching dictionary entry leads to multiple tissue properties directly via a look-up-table (LUT) operation. More specifically, the pre-computed dictionary is composed of a number of magnetic responses for a variety of tissues characterized by the values of their intrinsic properties, such as T1, T2 relaxation times, etc. In this way, each dictionary entry is associated with a specific tissue and its properties. Thus, once the best matching entry (i.e. most correlated element with respect to the enquiry in terms of their inner product, a.k.a. $\ell_2$ distance) is found, it directly leads to multiple properties simultaneously through a LUT. However, high computation and storage burden makes DM-based approaches prohibitively time-consuming and memory-consuming when the number of types and values of tissue properties increases, because the size of the dictionary and lookup table increases exponentially accordingly. To alleviate such drawbacks, other model-based MRF approaches were proposed to improve the speed, incorporate additional useful priors, reduce the computational complexity [6, 11, 38, 46, 47, 58].

Model-based MRF methods tend to suffer from burdens of storing the (enormous) dictionary and computational overhead. To address these shortcomings, learning-based approaches have been proposed for fast MRF by eliminating/replacing the dictionary by a compact neural network [8, 9, 15, 16, 18, 20, 25, 50, 55, 57]. In particular, motivated by the success of deep learning [19, 36] in a number of tasks [13, 17, 21, 22, 24, 32, 34], there is an emerging trend [8, 9, 15, 16, 18, 20, 25, 50, 55, 57] that suggests to use a trained neural network as an alternative substitute for the MRF dictionary and LUT so that the time-consuming dictionary matching operation can be eliminated and replaced by an efficient inference through a trained network. In fact, a well designed and tuned neural network is capable of approximating arbitrarily complex functions, therefore should also be able to approximate the response-to-property mapping function. Although these learning-based MRF approaches demonstrated better performance in terms of speed and accuracy in comparison with model-based variants, they also exhibit some limitations, such as degraded robustness and generalizability to various data shifts and out-of-distribution (OOD) data samples. Moreover, most of them are designed empirically without taking into account the MRI physics underlying the imaging process.

## 2.2 PHYSICS-INFORMED LEARNING

Another highly related line of research is physics-informed machine learning, a.k.a. model-based deep learning [29, 49, 54, 56] where a physics-based prior is embedded in the learning process. One of the typical examples is physics-informed neural networks (PINNs) [51] which are usually trained with both a data fitting loss and a regularization loss induced by physical priors, such as PDE, sparsity, low-rank, etc. Combining a physics prior in designing neural networks was shown to be beneficial to a broad range of applications [5, 14, 26, 44, 48]. PINNs are able to generalize well with a limited number of training data. In a similar spirit, we aim to incorporate the physics model that describes the MRI process into our neural networks to improve its data efficiency and generalization.

Physics-informed / model-based learning also appears in medical imaging domain [2, 7, 8, 27, 61, 62]. This provides a promising path to incorporate domain/expert knowledge with learned priors, and further combine the benefits of model-based methods with learning-based methods to achieve improved interpretability, transparency, and robustness. [8] proposes to incorporate the Bloch dynamics in an encoder-decoder framework. However, the decoder in their work, is actually a learned network rather than an MR physics model. Actually, the decoder just simulates and approximates the Bloch equations, instead of leveraging the exact physics model as in our case. Therefore, [8] is less physics-oriented than our proposed approach in essence. Reference [28, 37, 53] focus on how to design better scan parameters to obtain RF pulse sequences with higher sensitivity and effectiveness. To this end, it is formulated as an optimization problem that involves automatic differentiation of the Bloch equation. We admit that RF sequence optimization is very important for MRF. However, designing more sensitive and effective RF sequences is beyond the scope of our manuscript, as our goal is to improve the performance, efficiency, and robustness of mapping magnetic response to

tissue properties. [33] applies the analytic form of Bloch equations to generate synthetic qualitative MR images. Even though they include a physics model in the training procedure, they only consider one conventional RF pulse sequence setting (a fixed TE/TR/FA value) where a simple analytic form of Bloch equations exists. In our work, various RF settings are applied where complex non-analytic forms of Bloch equations are exploited, and therefore the encoder could capture generalizable representations from diverse magnetic responses, for example, from different hospitals. In addition, the application scenario of their work is in the synthesis of weighted MR contrasts, i.e. qualitative MRI domain, whereas our work focuses on quantitative MRI, in particular MRF problem. [42] proposes a supervised learning framework to generate MR images based on differentiable Bloch equation simulations from tissue parameters to be close to the target image contrast, such as shorter scan times, or higher SNR. [52] implements the fast EPG code and optimizes tissue parameters by repeating back propagation several times, leading to a much longer inference time than learning-based MRF methods. Our method, however, has the same inference time as deep neural network-based MRF methods since the Bloch equations are used only in the training time to provide additional useful feedback for training the encoder.

## 2.3 MRF PROBLEM SETTING

**Generation model (Bloch equations)** The data generation model of producing magnetic responses from tissue properties is based on the MRI physics model formulated by the Bloch equations [4]. Given the RF pulse sequence whose parameter setting $\Phi = \{FA, TR, TE\}$ consists of flip angles $FA \in \mathbb{C}^L$, repetition times $TR \in \mathbb{R}^L$ and echo times $TE \in \mathbb{R}^L$ across $L$ time points, the temporal signal evolution $X_n \in \mathbb{C}^L$ for each individual voxel $n$ is associated with tissue properties such as $\Theta_n = \{T1_n, T2_n\}$, through the Bloch differential equations $\mathcal{B}(\Theta_n) : \mathbb{R}^p \to \mathbb{C}^L, p = 2$.

$$X_n = \mathcal{B}(\Theta_n | \Phi) \quad \forall n \in 1, \dots, N$$

The Bloch equations $\mathcal{B}$, shown as Equation equation 1, are composed of a set of linear ordinary differential equations that represent a mapping from per-voxel intrinsic tissue properties to the corresponding temporal signal evolution that records the magnetization response of proton dipoles to dynamic excitations induced by the RF sequence.

$$\frac{d\vec{M}}{dt} = \vec{M} \times \gamma \vec{B} - \frac{\vec{M}_{xy}}{T2} - \frac{\vec{M}_z - \vec{M}_0}{T1} = \vec{M} \times \gamma \vec{B} - \begin{bmatrix} M_x/T2 \\ M_y/T2 \\ (M_z - M_0)/T1 \end{bmatrix} \tag{1}$$

where $\vec{M} = [M_x, M_y, M_z]^\top$ is magnetization with $\vec{M}_{xy}$ and $\vec{M}_z$ as the transverse and longitudinal components, respectively. $\vec{M}_0$ is the equilibrium magnetization; $\vec{B}$ is the magnetic field; $\gamma$ is the gyromagnetic ratio. Note that $X_n := \vec{M}_{xy}$, meaning only the transverse component of magnetization is read out and used to infer tissue parameters. (see Appendix for more details about Bloch equations and their differentiability.)

**Inverse model** Given the magnetic response $X_n \in \mathbb{C}^L$ for the $n$-th voxel, the inverse process aims to address an inverse problem that maps the response back to the corresponding tissue properties $\Theta_n$.

$$\Theta_n = g(X_n) \quad \forall n \in 1, \dots, N$$

where $g$ denotes the inverse mapping function. Note that, estimation of tissue properties $\Theta$ from magnetic responses $X$ requires long enough sequences $L > p$ to create unique signal evolutions that distinguish different tissues. Hence, the magnetic responses live on a low-dimensional (nonlinear) sub-manifold of $\mathbb{C}^L$.

## 3 PROPOSED METHOD

### 3.1 REPRESENTATION LEARNING FOR MRF

Here, we present a representation learning method to achieve fast and robust MRF. The proposed approach adopts a supervised encoder-decoder framework where the encoder predicts tissue properties from input signatures, while the decoder reconstructs the inputs signatures from the estimated

tissue properties based on a MRI physics model. We highlight that a sophisticated MRI physics model is tailored and exploited as the decoder which plays the role of causal regularization and help the encoder learn generalizable representations effectively. The rationale is based on the fact that representations that reflect the underlying data generation mechanism tend to exhibit stronger generalization to out-domain distribution, and the domain-specific physics based knowledge can help to reduce the solution space, thereby contributing to a better solution [31, 35, 45].

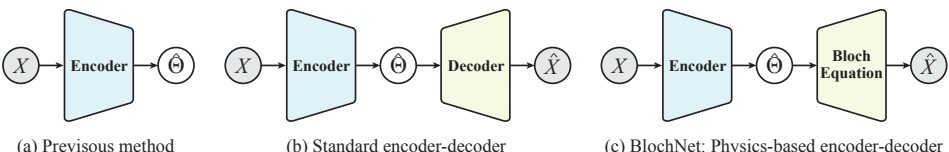

(a) Previsous method     (b) Standard encoder-decoder     (c) BlochNet: Physics-based encoder-decoder

Figure 2: Two baseline methods and our BlochNet. BlochNet exploits physics-based decoder for helping the encoder learn generalizable representation.

### 3.2 Encoder and Decoder

In the proposed approach, the encoder uses a three-layers fully connected neural network to predict T1, T2 parameters from fixed time-length input signatures. In contrast, the decoder leverages the Bloch equation based MRF physics model, acting as a physics-informed regularization, to perform supervised reconstruction to the input signatures from estimated tissue properties $\Theta$ and the given RF pulse $\Phi$.

Given an enquiry signature $X_n \in \mathbb{C}^L$ for the $n$-th voxel, the encoder $E$ solves an inverse problem and outputs predicted tissue properties $\hat{\Theta}_n = \{\hat{T1}_n, \hat{T2}_n\}$. [1] This operation nonlinearly maps the input signature from a high dimensional manifold to a low dimensional manifold.

$$\hat{\Theta}_n = \mathcal{E}(X_n) \quad \forall n \in 1, \ldots, N$$

Given the RF sequence settings $\Phi$ and the estimated tissue properties $\hat{\Theta}_n$, the decoder reconstructs the input signature via solving Bloch equations using extended phase graph (EPG) formalism [23, 59].

$$\hat{X}_n = \mathcal{B}(\hat{\Theta}_n | \Phi) \quad \forall n \in 1, \ldots, N$$

where $\mathcal{B}$ denotes the Bloch equation based decoder (see Appendix for more details).

### 3.3 Fast Bloch decoder based on efficient EPG

Since there is no general analytic solution to the Bloch equations, numerical solutions such as EPG formalization are often adopted (see Appendix for details about EPG). However, a significant limitation of the released EPG code [59] is its slow computation speed in solving the Bloch equations. To circumvent this, recurrent neural network [40] and generative adversarial networks [60] have been applied as surrogates for the Bloch equation. However, these require a lot of training data and still may generate inaccurate sequences on unseen tissue parameters and RF pulse settings due to overfitting. Instead, we adapt the EPG code [59] to achieve a much more efficient implementation, making it practical to use the exact MRI physics model as a decoder in the training procedure. A key change involves incorporating the torch jit package, and using batch-wise computation for the 3 Bloch stages, leading to 500 times faster generation of magnetic responses for 1,000 sequences on CPU. [2] More details can be found in the appendix.

---

[1] In our experiment setting, two properties including the longitudinal T1 and transverse T2 relaxation times, are simultaneously encoded for each voxel. This setting could be further extended to include other properties, e.g. proton density $\rho$, off-resonance frequencies, T2*, diffusion and perfusion.

[2] Our code, including the fast and end-to-end solvable EPG-Bloch code, is released on our GitHub repository at *[text left blank during peer review]*.

### 3.4 LOSS FUNCTION

The loss function consists of two parts: the mean squared error (MSE) between the ground truth and the predicted tissue properties, referred to as embedding loss, and the MSE between the input and the reconstructed signatures, referred to as reconstruction loss,

$$\mathcal{L} = \frac{1}{N} \sum_{n=1}^{N} \left( \frac{1}{2} \|\hat{\Theta}_n - \Theta_n\|_2^2 + \frac{1}{2} \|\hat{X}_n - X_n\|_2^2 \right)$$

## 4 EXPERIMENT RESULTS

In this section, we perform evaluation on the proposed method and conduct comparison with other state-of-the-art MRF methods. We evaluate the generalization performance of all models across different data distributions and different RF pulse sequences.

### 4.1 DATA SETTINGS

#### 4.1.1 SYNTHETIC DATA

The synthetic signature data $X$ is generated by solving the Bloch equations using the adapted fast EPG formulation for a set of ground-truth tissue properties $\Theta = \{T1, T2\}$, given the fixed RF pulse sequences with setting $\Phi = \{FA, TR, TE\}$. Specifically, the tissue properties $\Theta$ are combinatorial pairs of $T1$ values that range from 0 to 5000, and $T2$ values that range from 0 to 2500, following the settings used in [47, 55]. We also perform normalization for each signal evolution $X_n$ to remove the impact of the proton density which plays the role of a scale factor in front of the Bloch equations. This prevents the encoder from predicting tissue parameters based solely on magnitude of sequence instead of intrinsic properties of sequence. Every flip angle $FA$ was constrained to be in the interval 0 to 120 degree, and $TE, TR$ in second-scale. Note that predicted tissue properties $\hat{\Theta}_n$ are in logarithmic scale and they are reversed back to original scale before getting into the Bloch decoder.

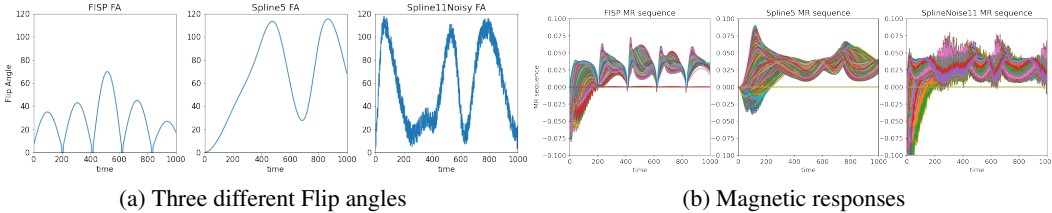

(a) Three different Flip angles         (b) Magnetic responses

Figure 3: Flip angles of three RF pulse sequences, including FISP, Spline5, Spline11Noisy, and corresponding magnetic responses generated using the Bloch equations for 16,384 data points(in different color).

#### 4.1.2 PHANTOM MRI DATA

We exploit the fuzzy version of the brain phantom of the BrainWeb Brain Database [10] to construct a set of realistic, high-resolution T1 maps and T2 maps as ground truth, which faithfully exhibit spatial distribution for different tissue compositions, including CSF, Grey matter, White matter, etc. Then, we select a few slices from each pair of T1/T2 phantom volume across 10 subjects. After removing non-brain parts, these slices are then vectorized as column vectors and stacked as a 2D matrix, leading to the ground truth tissue properties $\Theta \in \mathbb{R}^{N \times 2}$, where $N = 85,645$ denotes the total number of voxels from all slices. Given $\Theta$, signatures $X \in \mathcal{R}^{N \times L}$ are generated using the Bloch equation-based MRI physics model. (More details are provided in the appendix.)

#### 4.1.3 ANATOMICAL MRI DATA

The anatomical data is from [47, 55]. Specifically, brain MRI scans were first acquired from a healthy subject using GE Signa 3T HDXT scanner with Fast Imaging Employing Steady-state

Acquisition (FIESTA) and Spoiled Gradient Recalled Acquisition in Steady State (SPGR) at four different flip angles $(3°, 5°, 12°, 20°)$. Then corrections [39] are implemented, followed by using DESPOT1 and DESPOT2 algorithms [12] to obtain T1, T2 maps of size of $128 \times 128$ for reference, leading to $\Theta \in \mathbb{R}^{N \times 2}$ with $N = 7,499$ voxels after removing non-brain parts. Then the signatures $X$ are generated from the Bloch equations with FISP RF pulse sequence.

## 4.2 BASELINE METHODS

We compare our approach with 6 representative state-of-the-art MRF methods, including dictionary matching (DM) [43], Fully-connected deep neural network (FC) [9], Hybrid deep learning (HY-DRA) [55] as well as two auto-encoder methods with RNN encoder and RNN decoder(RNN-RNN) and FC encoder and FC decoder(FC-FC), respectively. The first three models are trained with only embedding loss, whereas auto-encoder models have an additional reconstruction loss for the decoder, apart from the embedding loss for the encoder. In constrast to the two auto-encoder models where the decoder is learned together with the encoder during training, our physics-based model exploits the Bloch equation based decoder and keeps the decoder fixed, while only the encoder is updated during training. The idea is to use the decoder as a causal regularization which gives reconstruction loss to guide the encoder to find more generalizable representations than baseline models.

| | Dictionary matching | FC | RNN | HYDRA | Autoencoder (FC-FC) | Autoencoder (RNN-RNN) | BlochNet (FC-Bloch) |
|---|---|---|---|---|---|---|---|
| Phantom data | 18.6652 | 0.0554 | 0.0486 | 0.1597 | 0.0519 | 0.0443 | **0.0409** |
| Anatomical data | 19.4883 | 0.0812 | 0.092 | 0.3088 | 0.0801 | 0.0889 | **0.0748** |

Table 1: Generalization performance across **different data distributions**: synthetic data for training while phantom (top row) and anatomical data (bottom row) for testing.

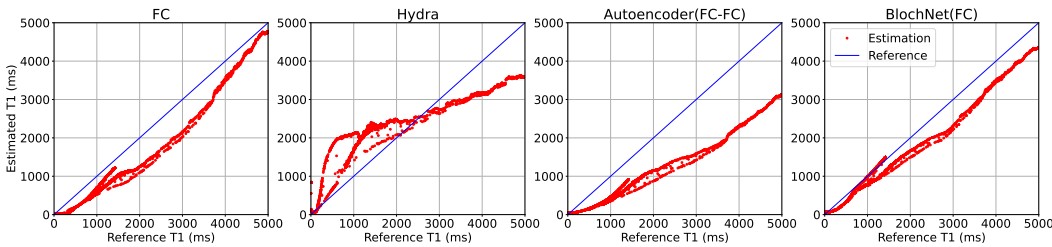

(a) T1 gold standard and predicted values for four models.

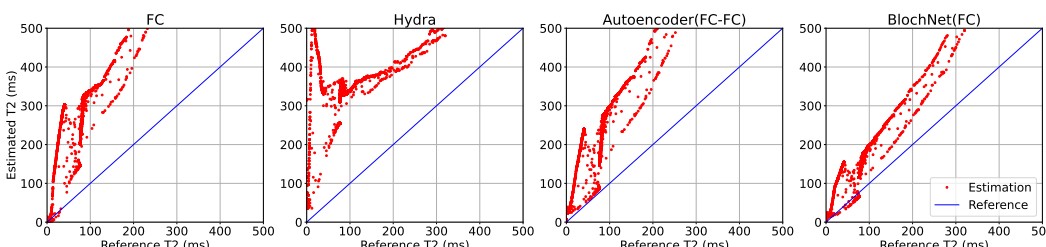

(b) T2 gold standard and predicted values for four models.

Figure 4: Comparison of the generalization performance across **different RF pulse sequences** for 4 models. Blue line: gold-standard. Red dots: predicted values for tissue properties T1 and T2.

| | Dictionary matching | FC | RNN | HYDRA | Autoencoder (FC-FC) | Autoencoder (RNN-RNN) | BlochNet (FC-Bloch) |
|---|---|---|---|---|---|---|---|
| Phantom data | 27.1955 | 0.8415 | 0.7973 | 0.3593 | 0.7018 | 0.6148 | **0.1574** |
| Anatomical data | 16.6151 | 0.7305 | 1.06603 | 0.695 | 0.4081 | 0.9385 | **0.2667** |

Table 2: Generalization performance across **different RF pulse sequences**: Spline5 and SplineNoisy11 in training, FISP in testing.

### 4.3 EXPERIMENTS OF EVALUATING GENERALIZATION PERFORMANCE

We evaluate the generalization performance of various models on two types of experiment settings: 1) across different data distributions, including synthetic, phantom and anatomical MRF data; 2) across different RF pulse sequences with different flip angles.

#### 4.3.1 GENERALIZATION ACROSS DIFFERENT DATA DISTRIBUTIONS

In practice, acquiring anatomical MRF data with ground truth T1, T2 values is time-consuming and expensive. Due to limited labeled anatomical data, it is common practice to use a large amount of synthetic data to train models to avoid potential overfitting, and then perform validation on anatomical data [6, 11, 18, 38, 46, 47, 55, 58]. Following the same routine, we perform model training on synthetic MRF data, followed by model testing on phantom and anatomical data, in order to evaluate the generalization performance of trained models across different data distributions. In this experiment, we perform model training on synthesized MRF data introduced in Section 4.1.1 for our model and baseline models, and then compare their performance on unseen phantom and anatomical data described in Section 4.1.2 and 4.1.3.

Table 1 includes the mean squared error (MSE) between the ground truth and predicted tissue properties for seven approaches on phantom and anatomical data. Note that these errors are computed in log-scale. As shown in the table, the dictionary-matching approach gave the worst performance, because the pre-computed dictionary and LUT did not cover the OOD data samples that could be quite different from the already contained dictionary entries. This caused poor inner product based matching results, and restricted the generalization.

Interestingly, the results show that the reconstruction loss provides benefits to between-data generalization for autoencoder models(FC-FC or RNN-RNN), in comparison with non-autoencoder models(FC or RNN), respectively, on both phantom and anatomical MRF data. Furthermore, our BlochNet outperforms all other models, indicating that reconstruction loss from the physics-based decoder has the best regularization effect that contributes to improved encoder training. In particular, when the physics-based decoder reconstructs input signatures from estimated tissue properties using Bloch equations, the corresponding reconstruction loss serves as additional guidance and constraint for the encoder to effectively capture the underlying anti-causal mapping. This in turn leads to better generalization performance.

Figure 5 shows the predicted tissue properties using various models on anatomical MRF data. All models perform well in the middle range of tissue properties and lead to small errors. However, each individual models show different prediction characteristics. Specifically, HYDRA, as shown in the third column of 5, suffers from a higher loss at the rim region of the brain, and leads to larger errors than other models. Autoencoder(FC-FC) model, as shown in the forth column, demonstrates a better prediction of T2 values than the non-autoencoder(FC) model in the second column. The proposed BlochNet outperforms other comparison models with the least prediction error and most stable performance across the whole range of both T1 and T2 values, as shown in the fifth column.

#### 4.3.2 GENERALIZATION ACROSS DIFFERENT RF PULSE SEQUENCES

In this experiment, we perform model training on one RF pulse sequence and evaluate the trained models on another different RF pulse sequence. Specifically, we adopted 3 different RF pulse sequences, including FISP [43], Spline5 [40], Spline11Noisy [40] with their flip angles shown in Figure 3a. (More details are provided in Appendix.) FISP is used exclusively in the testing stage, while Spline5 and Spline11Noisy are used exclusively in the training stage. Under such settings, the performance of our BlochNet and other six models is compared in Table 2.

Both Table 2 show that the embedding loss for all models are much higher than Table 1 regardless of phantom and anatomical MRF data. This is because the signatures generated with Spline5 and Spline11Noisy RF pulse sequences have notable difference from signatures for FISP, as shown in Figure 3b, which are particularly challenging cases. In spite of degraded performance for all models, the results clearly show the advantage of autoencoder (FC-FC or RNN-RNN) models over non-autoencoder models(FC or RNN), which confirms the benefits of incorporating an decoder to derive the reconstruction loss as additional regularization. Furthermore, the proposed BlochNet demonstrates significant gains over the competing methods in such challenging cases on both phan-

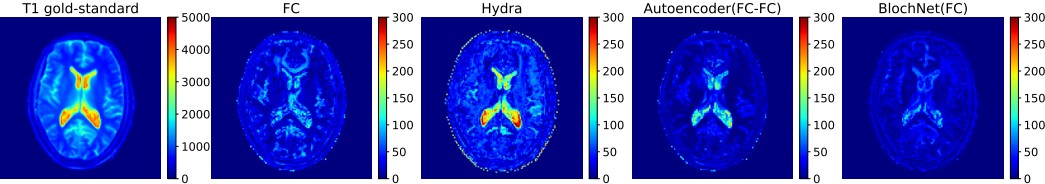

(a) Gold-standard T1 and errors between gold-standard and predicted T1 values of four models.

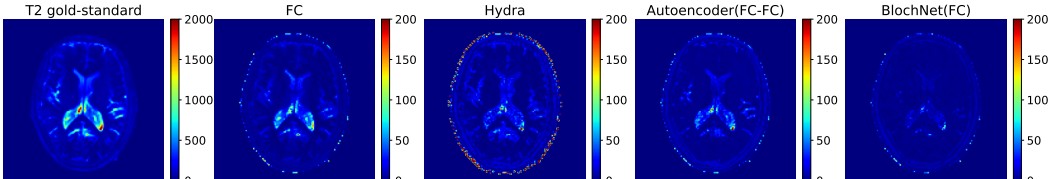

(b) Gold-standard T2 and errors between gold-standard and predicted T2 values of four models.

Figure 5: Generalization performance across different data distributions: data in training is synthetic while in testing is anatomical MRI for four different models.

tom and anatomical MRF data. For example, the reconstruction loss of 7 baseline models is larger than 0.3 while our BlochNet gives 0.1574 on phantom MRI data. In all these settings, our approach leads to at least 50% smaller loss than the best competing approach.

These differences in MSE losses are shown in Figure 4. FC model (left) makes poor predictions on both T1 and T2 values with high variance, predicting different values even when the sequence is generated from the same tissue properties. This happens when the model cannot infer tissue properties from input signatures generated from different combinations of T1 and T2 values. Autoencoder(FC-FC) model (third column) shows more aligned and better inferences with lower variance, but still has high deviation between predicted and gold-standard values. In comparison, our BlochNet outputs predictions that are closest to the gold-standard with the lowest error. This confirms the benefits of our physics-based decoder that guides the encoder to learn the underlying anti-causal mechanism effectively.

## 5 CONCLUSION

We propose BlochNet, a novel physics-informed learning model, to perform quantification of multiple tissue properties from their magnetic responses, a challenging ill-posed inverse problem in quantitative MRI. By integrating the encoder-decoder framework with Bloch equation based MRI physics, an additional reconstruction error resulting from the Bloch decoder can be exploited to regularize the training of the encoder, and accordingly allows the well-train encoder to achieve improved multi-property quantification performance. Experiments demonstrate that our method consistently outperforms competing methods with better robustness and generalizability.

In future work, we will consider the k-space subsampling circumstance and spatial information that may lead to faster and more efficient QMRI/MRF. We will also explore more varied RF settings, for example, time-varying TR, TE and FA, as it has been reported [3] that a RF pulse sequence with multiple pseudo-random parameters may improve the identifiability of resulted magnetic responses.

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
