# OpenReview forum: "Physics Model-based Autoencoding for Magnetic Resonance Fingerprinting"
_ICLR.cc/2023/Conference — Submitted to ICLR 2023_

### Official Review · Reviewer_oTWC · 2022-10-24

**Confidence:** 3
**Correctness:** 2
**Technical Novelty And Significance:** 2
**Empirical Novelty And Significance:** 2
**Recommendation:** 3

**Clarity, Quality, Novelty And Reproducibility:**

As stated before, there are some missing references. The authors should discuss how the proposed approach is different from state-of-the-art.
The authors aim to provide source code for reproducibility purpose.

Some minor issues:
•	When claiming superior runtimes, computation times should be reported.
•	In Figure 3, the subfigures appear slightly shifted, axis labels and color legends are missing, and some text could be sharper.
•	Are the highlighted scores of Table 1 significantly better than the baseline autoencoder approaches? The reported errors seem very close.


**Strength And Weaknesses:**

Strengths:
1.	The authors contribute to the field of MR fingerprinting by presenting a network that builds on MR physics models and demonstrates a promising generalization performance.
2.	Providing a fast implementation for solving Bloch equations appears to be an impactful contribution to the research community. Sharing the GitHub code improves the reproducibility of their work.
3.	A thorough evaluation of the BlochNet and multiple baseline comparisons was performed on three datasets.
4.	They present appealing visualizations

Weaknesses:
1.	The literature review appears incomplete. An example is the work of Kim et al. who proposed a Bloch equation-based decoder in ‘Fat-saturated image generation from multi-contrast MRIs using generative adversarial networks with Bloch equation-based autoencoder regularization’ published in Medical Image Analysis, 2021. The authors should also check the recent contributions by Jesse Hamilton, especially “Deep learning reconstruction for cardiac magnetic resonance fingerprinting T1 and T2 mapping”, MRM, 2021. This brings questions to the stated novelty of the proposed work.
2.	The discussion of causal and anti-causal mapping appears contextless
3.	There are major grammar and format errors that need to be addressed (e.g. wordy sentences, misplaced citations, inconsistencies in the use of capital and small initial letters as well as blank spaces)


**Summary Of The Paper:**

This paper presents a deep learning based reconstruction for Magnetic Resonance Fingerprinting (MRF). The proposed approach based on a AutoEncoder scheme, where the decoder block is inspired by the physics. The so-called BlochNet allows to integrated priori knowledge on the physics driving the generation of MRI, and ensuring that the latent space respects the Bloch equations.

**Summary Of The Review:**

The proposed approach seems interesting with the use of prior physics to drive the training of an autoencoder. However, the authors should discuss how their proposed approach is different form missing references, and compare their technique with this state of the art.

---

> ### Author Response · Authors · 2022-11-15
>
> We thank the reviewer for their careful reading of our work and the critical evaluation of our contribution. We have considered their input and have uploaded a revised paper.
>
> ### On incomplete literature review
> We have enriched the literature review noted by the reviewer and incorporated more relevant references. Please refer to the updated manuscript in the Related work section.
> Below, we clarify the differences between our work and the newly-added reference. First, [1] applies a commonly-used deep learning-based MRF idea to a specific cardiac situation and takes into account its unique characteristics. We highlight that there is no MRI physics model exploited in their work, which is the main difference from our work. The core idea of replacing the MRF dictionary and associated dictionary matching operation by a neural network is in line with other deep learning MRF approaches, such as [3,4]. We clearly showed that our BlochNet had a better performance in OOD setting than these purely data-driven approaches in Section 4.
>
> Second, [2] applies the analytic form of the Bloch equation to generate synthetic qualitative MR images. Even though they include a physics model in the training procedure, they only consider one conventional RF pulse sequence setting (a fixed TE/TR/FA value) where a simple analytic form of Bloch equations exists. In our work, various RF settings are applied where complex non-analytic forms of Bloch equations are exploited, and therefore the encoder could capture generalizable representations from diverse magnetic responses, for example, from different hospitals. Please refer to Table 2. Also, the application scenario of their work is in the synthesis of weighted MR contrasts, i.e. qualitative MRI domain, whereas our work focuses on quantitative MRI, in particular MRF problem.
>
>
>
> * [1] Hamilton, J. I., Currey, D., Rajagopalan, S., & Seiberlich, N. (2021). Deep learning reconstruction for cardiac magnetic resonance fingerprinting T1 and T2 mapping. Magnetic Resonance in Medicine, 85(4), 2127-2135.
> * [2] Kim, S., Jang, H., Hong, S., Hong, Y. S., Bae, W. C., Kim, S., & Hwang, D. (2021). Fat-saturated image generation from multi-contrast MRIs using generative adversarial networks with Bloch equation-based autoencoder regularization. Medical Image Analysis, 73, 102198.
> * [3] Song, P., Eldar, Y. C., Mazor, G., & Rodrigues, M. R. (2019). HYDRA: Hybrid deep magnetic resonance fingerprinting. Medical physics, 46(11), 4951-4969.
> * [4] Cohen, O., Zhu, B., & Rosen, M. S. (2018). MR fingerprinting deep reconstruction network (DRONE). Magnetic resonance in medicine, 80(3), 885-894.
>
>
>
> ### On unclear explanations surrounding causal and anticausal
>
> <This is the same response as reviewer 1>
> We agree with the reviewer that using concepts of ‘causal’ and ‘anticausal’ could make our point hard to read. We updated our paper to deliver our point in a different way in the main paper.
> The reason why we applied the concept of causal and anticausal was to explain why getting additional feedback from the MRI physics model assists the encoder to learn features generalized to out-of-distribution data samples. The key point lies in the fact that the MRI physics model provides additional useful feedback for training the encoder because mapping predicted tissue parameters back to magnetic responses as a comparison with the original input provides another perspective to evaluate and update the encoder. In the updated paper, we re-phrasing it in a similar spirit of physics-informed neural networks
>
>
> ### On grammar and format errors
> We polished our new version of the manuscript and corrected grammar and format errors.
>
> ### On minor issues
>
> About the scores of Table 1, we verify that the proposed model exhibits improved generalization performance from synthetic to real MRF data in two different out-of-distribution (OOD) settings, where the test data is generated from seen and unseen RF pulses in the first and second settings, respectively.
>
> In the first setting, looking at the shape of the input is enough to achieve high performance, therefore all approaches show good results on this easy task. On the other hand, in the second setting, the results show that our method works better with big loss differences in this hard task where capturing generalizable features is required.
>
> Thank the reviewer for pointing out issues in the Figure 3, and we revised it.

---

### Official Review · Reviewer_F4Mm · 2022-10-25

**Confidence:** 4
**Clarity, Quality, Novelty And Reproducibility:** The  novelty of this paper is quite l…
**Correctness:** 2
**Technical Novelty And Significance:** 2
**Empirical Novelty And Significance:** 2
**Recommendation:** 3

**Strength And Weaknesses:**

Strengths:
-  Including physics of the problem is  important for the medical imaging applications.
- The paper experimental results generally align with earlier reports on the importance of physics-based models (Monga et al, SPM 2021)

Weaknesses:

1) Physics-based models are a well-studied field of work across different domains. The physics-based models are known to be more robust and generalizable compared to data-driven/learning-based approaches (Monga et al, SPM 2021). In fact, for MRI reconstruction tasks such as fastMRI challenge (Muckley et al, TMI 2021), physics-based models have been consistently preferred over data-driven/learning-based approaches. Thus, including the physics of the problem in the network architecture is not a novel contribution, but a straightforward extension of MRF.

2) The generalization performance is questionable. While physics-based model improves the performance over learning-based models, they do not necessarily address the out-of-distribution challenges

    a) Based on the findings in Table 1, auto-encoder models such as RNN-RNN do not suffer from the out-of-distribution challenge.  This somehow contradicts the motivation of the study which introduces physics-based modeling to improve the generalization performance. More concrete experiments must be done to show a clear advantage when facing different data distributions.

   b) Figure 4 results show that BlochNet improves compared to other methods in presence of different RF-Pulse sequences, however, compared to reference it is still far off. Thus, it does not really solve the OOD challenge.

3) Including the physics of the problem tends to have a counter impact on the inference time. A comparative study comparing the inference time is important as inference time plays an important role in medical imaging applications.

4) Please revise the writing as there are repetitive statements (see the last paragraph of the section 2.2)


**Summary Of The Paper:**

This paper introduces a physics-based approach, named BlochNet, for magnetic resonance fingerprinting. Unlike existent works BlochNet directly incorporates Bloch equations in the decoder for improved generalizability. The experiments show improvement compared to conventional and learning-based approaches.

**Summary Of The Review:**

Overall, the novelty is quite limited. motivation and experimental result does not align well. The novelty and experiments need to be improved for ICLR.

---

> ### Author Response · Authors · 2022-11-15
>
> We thank the reviewer for taking the time to read our paper and for their valuable suggestions to improve the clarity of our work. We have incorporated the feedback into the updated paper.
>
> ### On the discussion of the novelty
> We are aware that physics-based machine learning models are attracting an increasing amount of attention and have been applied to a variety of applications, as shown in the recommended paper (Monga, et al SPM 21 "Algorithm unrolling: Interpretable, efficient deep learning for signal and image processing"). These works focus on novel deep neural network designs by unrolling iterative optimization algorithms that are usually developed for solving particular problems. We highlight that the same physics model can be exploited in different ways to demonstrate different benefits. Therefore, newly-developed physics-based models have their own contributions to new application scenarios.
> Likewise, even though the concept of applying physics-based models is not new, there have been no approaches to include the Bloch equation in the training procedure for guiding the encoder to capture generalizable features in the MR fingerprinting problem.
> Also, we incorporated new creative work into it. For example, MR fingerprinting is different from qualitative/contrast/weighted MRI, and its characteristics need to be taken into account when applying physics-based modeling technology, such as how to balance different tissue parameters, how to ensure the target tissue properties are identifiable from input magnetic responses, what influence the model has on the reconstruction stability, etc. which are new issues that require particular attention in comparison with conventional qualitative/weighted MRI.
>
>
> ### On addressing OOD challenges
>
> In our work, we verify that the proposed model exhibits improved generalization performance from synthetic to real MRF data in two different out-of-distribution (OOD) settings, where the test data is generated from seen and unseen RF pulses in the first and second settings, respectively.
>
> In the first setting, looking at the shape of the input is enough to achieve high performance, therefore all approaches show good results on this easy task. On the other hand, in the second setting, the results show that our method works better with big loss differences in this hard task where capturing generalizable features is required.
>
> We agree that our work couldn’t totally solve the second OOD setting. However, it is still meaningful to achieve way better results compared to other approaches. We think our work is the first step towards the generalization problem in the second OOD setting, which has been neglected even though it is practical, and we hope to enhance performance for future work.
>
>
> ### On the inference time
>
> We agree that inference time is important in medical imaging applications. During the training time, the encoder is trained with both prediction loss and reconstruction loss using our physics-based decoder, therefore training time is longer than learning-based baseline approaches (e.g., simple FC, CNN, or autoencoder without physics-based decoder). However, in the inference time, only the trained encoder is used to predict tissue parameters in MRF, thus our method which uses FC as the encoder has the same inference time as baseline methods.
>
> ### On repetitive statements
> We thank the reviewer for pointing out this, and we updated our paper after revising them.

---

### Official Review · Reviewer_29qt · 2022-10-30

**Confidence:** 4
**Correctness:** 3
**Technical Novelty And Significance:** 4
**Empirical Novelty And Significance:** 4
**Recommendation:** 3

**Clarity, Quality, Novelty And Reproducibility:**

The manuscript does not describe the procedure of either the EPG-based decoder, or the encoder. Since this is primarily a Machine Learning venue, explaining how the EPG method can be used to form a differentiable forward model in a reasonable amount of time with reasonable accuracy is, in my opinion, the strongest point of the paper. While the math can be relegated to the appendix somewhat, what it's skipping/making assumptions about and how the outcome in A.2 is exactly what we wanted to find would be very helpful to the reader (is it what we wanted to find? seems to be a k-space representation only in space, so we need this also across time?). A.5 includes computational details, but we need methodological details.

A consistent description of the encoder is necessary. Does this encoder take in more than a single voxel? Is it performed in k-space? (Coil correction?) Is it conditioned on the waveforms? How does it generalize well without knowing about the sequence?

Details of the phantom would be a nice inclusion also in the appendix, for those wanting to reproduce the lab results.

Additional comments:
* Should MSE be used without reweighting T1 and T2? These aren't on the same scale as far as I am aware.
* The description of the method as an "auto-encoder" is somewhat misleading for deep learning readers; this is a supervised method, while most auto-encoders only have priors/regularization and reconstruction loss. I think this is minor, but it might clarify questions here by renaming the paper somehow.

**Strength And Weaknesses:**

Strengths:
* This method demonstrates an excellent example of physical models being incorporated into deep architectures, and, conversely, deep architectures informing the estimation of real physical properties.
* The experimental results are promising, especially in the case of cross-sequence learning.
* This, in my opinion, is an excellent use case for deep learning in MRI, since the standard fingerprinting already uses a dictionary method (i.e., is already as a field conditioned on some form of learning), and since the inverse problem here is relatively stiff for some parameters/playbacks.

Weaknesses (extended comments in Clarity section):
* The authors do not describe the EPG method of reconstruction or its use. Even considering appendix A.1/A.2, usage is opaque. Since this portion is describing a critical portion of the method, care should taken so that this entire section is as explicit as possible.
* The same comment as above, but for the RNN(?) or FC network(?) used as an encoder. This section is completely opaque, except for the fact that some form of learning on the sequence of measurements has occurred. Moreover, section 3.0.2. implies that an RNN is used on the readout timeseries (of fixed length L), while Section 4 refers to BlochNet as (FC-Bloch).

**Summary Of The Paper:**

The authors propose a "auto-encoder" for MR Fingerprinting, which is an imaging method in magnetic resonance that captures multiple tissue properties at once (opposed to the "standard" T1 or T2 imaging, which captures a single tissue property, or their weighted variants T1w and T2w which only capture relative weightings of that property). The standard physical model taking these properties to the observed signal are the Bloch Equations, a system of ODEs.

The authors propose to learn an encoder, which (optimally) reconstructs the tissue properties from the timeseries observed measurements, and use the extended phase graph (EPG) method to approximate the forward model (Bloch Eq.s).

They then present results of various baselines (the standard dictionary method, and several deep architectures) on both a phantom and anatomical data (human brain).

**Summary Of The Review:**

I think this work is significant and novel, and important for the deep-learning/MRI sub-field, but I think it requires a lot more description of the exact construction of the deep learning method to be publication quality and/or reproducible. It is a good idea, and has the pieces of a good manuscript, but missing details in section 3 and inconsistency with section 4 are strong negatives.

Given a revision fixing these problems (or an explanation of what I have missed/misunderstood), I am willing to change my recommendation.

---

> ### Author Response · Authors · 2022-11-15
>
> We thank the reviewer for their careful reading of our paper and for their suggestions that have led to an improved manuscript.
>
> ### On the description of the Bloch equation model and EPG method
>
> We thank the reviewer for the suggestion. Appendix A.1 and A.2 include more detailed descriptions to the Bloch equation model and EPG method.
>
> EPG is an efficient way to compute the Bloch equations in the Fourier domain. So the differentiable property lies in the Bloch equation formulated in equations in Section 3.3 and A.1, where the magnetism M is formalized in an ODE form. Under some conditions, typically simple RF sequences and with assumed steady-state, there exist analytic (closed-form) solutions, e.g. M = M_0 (1- exp(-TR/T1) ) exp(-TE/T2) for spin-echo pulse sequence. Then, it is apparent that M is differentiable with respect to tissue properties T1, and T2. However, for a complex pulse sequence, analytical solutions are hard to obtain due to spin history effects at unsteady states and system imperfections. For example, in MR fingerprinting, various sequence components are varied in a pseudo-random pattern. In such cases, the solutions are also numerically differentiable with respect to T1 and T2.
>
> ### On the encoders
>
> We agree that there was confusion about the encoder in the original paper. Thanks to the reviewer for pointing this out. We have updated it to clarify. Below, we answer the reviewer’s questions about the details of the encoder point-by-point. We plan to incorporate those points for future work.
> * About the structure of the encoder, we only include the experiment results of FC as the encoder since we used a fixed time length MR sequence.
> * About the input shape, the encoder takes a single voxel with a time length L as mentioned in Section 4.1.
> * About k-space, we did not consider the k-space subsampling process in the current version, mentioned in future work in the Conclusion section.
> * About the waveforms conditioning, the decoder, i i.e. Bloch equation is conditioned on RF settings, including gradient waveform, TR, TE, FA, as they are needed to generate magnetic response for tissue properties. However, the encoder has nothing to do with RF settings and therefore is not conditioned on gradient waveform. This is because the input (magnetic response) already contains the information about RF. Please refer to the equations in Section 3.2.
> * Also, we added the details of the phantom data in the Appendix.
>
> ### On explanation about how the BlochNet generalizes well without knowing about the sequence
>
> We thank the reviewer for raising interesting and important questions. As shown in Figure 3, the shapes of the sequences are different with different flip angles. However, the BlochNet shows better tissue parameter prediction on the unseen magnetic responses generated from unseen scan parameters, such as flip angles.
> We conjecture that the reason is related to the MRI physics model-based decoder which has access to new RF pulse sequences and thereby provides a path for the encoder to exploit new scan parameters implicitly. (Suggestion: We may conduct an experiment where during training and testing, we always fix the RF pulse sequence including TR, TE, FA when inputting them to the Bloch decoder, then verify the performance on unseen magnetic responses that are generated from different RF pulse sequences.)
>
> ### On the additional comments
>
> On the reweighting of T1 and T2, thank you for the comment. We added a description of the data scale and normalization in Section 4.1. For T1 and T2 values, the predicted tissue parameters are in log-scale while the ground-truth values are not. If by “reweight” the reviewer means changing scales of tissue parameters, then they are correct. However, if they mean normalizing tissue parameters, then no we didn’t.
> On the additional normalization, we normalized the input temporal MR sequence to remove the effect of PD, which prevents the model from predicting tissue parameters simply based on the magnitude of a sequence instead of its intrinsic properties. We have included this in the updated manuscript.
> On the usage of “auto-encoder” and renaming it, we agree with the reviewer’s worry about misunderstanding our method by using the wrong name, thank you. Since we use both supervised MSE loss and reconstruction loss, naming it as “auto-encoder” can be inappropriate. We have changed it to “encoder-decoder” and updated it in the revised paper.

---

### Official Review · Reviewer_qrEB · 2022-11-01

**Confidence:** 5
**Correctness:** 3
**Technical Novelty And Significance:** 1
**Empirical Novelty And Significance:** 1
**Recommendation:** 3

**Clarity, Quality, Novelty And Reproducibility:**

The methodology is clear but the novelty is low. Though not provided, I expect the results to be reproducible.


**Strength And Weaknesses:**

Strengths:
- The paper defined the problem well and makes the benefit of QMRI clear.
- The paper proposes a solution to directly incorporating the Bloch equations into the reconstruction

Weaknesses:
- The paper incorrectly claims to be the first to incorporate the Bloch equations into the decoder. This is simply not true. See for example, [1-4]. In addition, the paper is not the first to accelerate EPG to make it usable as a decoder. See for example, [5]. It is unclear how the paper differs from these works, especially with respect to [1] and [5], except that these works can also be used to optimize the pulse sequence. The authors will need to explain how their approach is fundamentally different than these existing works.

[1] Philip K Lee, Lauren E Watkins, Timothy I Anderson, Guido Buonincontri, Brian A Hargreaves, Flexible and efficient optimization of quantitative sequences using automatic differentiation of Bloch simulations, Magnetic resonance in medicine 2019
[2] Beomgu Kang, Byungjai Kim, HyunWook Park, Hye-Young Heo, Learning-based optimization of acquisition schedule for magnetization transfer contrast MR fingerprinting, NMR in Biomedicine 2021
[3] A. Loktyushin, K. Herz, N. Dang, F. Glang, A. Deshmane, S. Weinmüller, A. Doerfler, B. Schölkopf, K. Scheffler, M. Zaiss, MRzero - Automated discovery of MRI sequences using supervised learning, Magnetic resonance in medicine 2021
[4] Evan Scope Crafts, Hengfa Lu, Huihui Ye, Lawrence L. Wald, Bo Zhao, An efficient approach to optimal experimental design for magnetic resonance fingerprinting with B-splines, Magnetic resonance in medicine 2022
[5] Somnath Rakshit, Ke Wang, and Jonathan I Tamir, A GPU-accelerated Extended Phase Graph Algorithm for differentiable optimization and learning, ISMRM 2021

- The paper throws around a lot of jargon surrounding causal and anti-causal mapping, but this makes it harder to read and interpret.
- The paper is heavy in exposition but not heavy in description. This makes it difficult to discern the contribution vs. background material.


**Summary Of The Paper:**

The authors propose BlochNet, an approach to reconstructing magnetic resonance fingerprinting (MRF) data. The approach takes the input images, solves for the parameter maps through an inverse problem, and then re-encodes the parameters into multi-contrast images. The encoder is implemented using a neural network and the decoder is implemented using the extended phase graph algorithm. Comparisons are made to other MRF reconstruction techniques across different RF pulse schedule.


**Summary Of The Review:**

The authors propose an auto encoder for MRF based on NN (encoder) + Bloch equations (decoder). Though powerful, the approach is not new and the paper does not provide any novelty.

---

> ### Author Response · Authors · 2022-11-15
>
> We thank the reviewer for their careful reading of our paper and for their suggestions on related works about applying EPG as a decoder. We answer the reviewer’s questions below and hope that we adequately clarify the points raised.
>
> ### On incomplete literature review
>
> The literature review in our original version of the manuscript focused primarily on MRF related work, with little attention to other MRI tasks. This resulted in some missing references pointed out by the reviewer. We thank the reviewer for the pointers to relevant work, which have been incorporated into our updated version.
>
> References 1 and 2 focus on how to design better scan parameters to obtain RF pulse sequences with higher sensitivity and effectiveness. To this end, it is formulated as an optimization problem that involves automatic differentiation of the Bloch equation. We admit that RF sequence optimization is very important for MRF. However, designing more sensitive and effective RF sequences is beyond the scope of our manuscript, as our goal is to improve the performance, efficiency, and robustness of mapping magnetic response to tissue properties. Therefore, the RF sequence adopted in our current work is the commonly-used FISP sequence, without considering the sequence optimization problem. We have cited the recommended reference 1 in our updated manuscript and will consider the sequence optimization problem in our future work.
>
> Here are the quoted sentences explaining the differences between our work and [1],[5] for convenience: “Reference [1] focus on how to design better scan parameters to obtain RF pulse sequences with higher sensitivity and effectiveness. To this end, it is formulated as an optimization problem that involves automatic differentiation of the Bloch equation. We admit that RF sequence optimization is very important for MRF. However, designing more sensitive and effective RF sequences is beyond the scope of our manuscript, as our goal is to improve the performance, efficiency, and robustness of mapping magnetic response to tissue properties. [5] implements the fast EPG code and optimizes tissue parameters by repeating back propagation several times, leading to a much longer inference time than learning-based MRF methods. Our method, however, has the same inference time as deep neural network-based MRF methods since the Bloch equations are used only in the training time to provide additional useful feedback for training the encoder.”
>
> ### On unclear explanations surrounding causal and anticausal
>
> We agree with the reviewer that using concepts of ‘causal’ and ‘anticausal’ could make our point hard to read. We updated our paper to deliver our point in a different way in the main paper.
> The reason why we applied the concept of causal and anticausal was to explain why getting additional feedback from the MRI physics model assists the encoder to learn features generalized to out-of-distribution data samples. The key point lies in the fact that the MRI physics model provides additional useful feedback for training the encoder because mapping predicted tissue parameters back to magnetic responses as a comparison with the original input provides another perspective to evaluate and update the encoder. In the updated paper, we re-phrasing it in a similar spirit of physics-informed neural networks
>
> ### On lack of description
>
> We thank the reviewer for pointing out that the description is not enough. In the revised version, we enhanced the description of the encoder (Section 3.2), Bloch equations (Section 2.3, Appendix A.1, A.2), and data (Section 4.1, Appendix A.4).

---

### Author Response · Authors · 2022-11-15
**Response Summary**

We thank the reviewers for their time and effort to provide helpful and constructive feedback and comments. We appreciate that reviewers acknowledged the importance of integrating MRI physics with machine learning and applying this such idea to QMRI / MR fingerprinting. We have meticulously revised our manuscript to incorporate each suggestion, and carefully addressed each question, doubt and concern raised by reviewers. The main revisions include:
 * (About our novelty) Our BlochNet is the first approach to incorporate physics-informed learning to solve MRF problems, achieving state-of-the-art generalization in practical out-of-distribution settings where MR responses are generated from unseen RF sequences with fast inference time.
* (R1, R4) Incorporated additional references, including recommended articles, with dicussion to cover related work more comprehensively;
* (R1, R4) Been more clear and specific in our claim that “we integrate the encoder-decoder framework with Bloch equation based MRI physics to achieve better generalization for MR fingerprinting”, to emphasize the specific application scenario.
* (R2, R3) Clarified the proposed model and related technical material in more detail, e.g. how the EPG-based MRI physics model is used in the training of our BlochNet and other competing models and how a physics prior is used in training neural networks.
* (R2, R4) Polished the language and description.

---

### Decision · Program_Chairs · 2023-01-20

**Decision:**

Reject

**Justification For Why Not Higher Score:**

All four reviewers recommend rejecting the paper for the following reasons: R1 and R3 for it's lack of novelty, R2 and R3 for not being publication-ready, and R4 for a lack of comparison with the state-of-the-art.
I agree with the reviewers that the paper does not provide a sufficient contribution to the literature to justify acceptance, and is not ready for publication, and therefore recommend rejecting the paper.


**Justification For Why Not Lower Score:**

N/A

**Metareview: Summary, Strengths And Weaknesses:**

The paper proposes a physics-based appraoch for magnetic resonance fingerprinting data.

Strength: The paper studies a relevant problem, and the approach of incorporating the physics (the Bloch equations) into the reconstruction algorithm is very sensible and works well.

Weaknesses: The methods is very sensible, but variants of it have been studied an proposed already. Moreover, the reviewers pointed out a variety of issues (such as a lack of comparison to related work, and issues on the clarity/writing).